REGISTERED REPORT PROTOCOL

# Developing a protocol for adapting multimedia patient-reported outcomes measures for low literacy patients

**Chao Long** [1,2]*, **Laura K. Beres**[3], **Albert W. Wu**[4], **Aviram M. Giladi**[1]

**1** Curtis National Hand Center, MedStar Union Memorial Hospital, Baltimore, Maryland, United States of America, **2** Department of Plastic and Reconstructive Surgery, Johns Hopkins Medicine, Baltimore, Maryland, United States of America, **3** Department of International Health, Johns Hopkins Bloomberg School of Public Health, Baltimore, Maryland, United States of America, **4** Department of Health Policy and Management, Johns Hopkins Bloomberg School of Public Health, Baltimore, Maryland, United States of America

* chaolong@jhu.edu

## Abstract

### Background

Self-administration of patient-reported outcomes measures (PROMs) by patients with low literacy is a methodologic and implementation challenge. There is an increasing emphasis on patient-centered care and wider adoption of PROMs to understand outcomes and measure healthcare quality. However, there is a risk that the use of PROMs could perpetuate health disparities unless they are implemented in an inclusive fashion. We present a protocol to adapt validated, text-based PROMs to a multimedia format (mPROMs) to optimize self-administration in populations with limited literacy. We describe the processes used to develop the protocol and the planned protocol implementation.

### Methods/Design

Our study protocol development was guided by the International Quality of Life Assessment (IQOLA) protocol for translating and culturally adapting PROMs to different languages. We used the main components of IQOLA's protocol to generate a conceptual framework to guide development of a Multimedia Adaptation Protocol (MAP). The MAP, which incorporates human-centered design (HCD) and takes a community-engaged research approach, includes four stages: forward adaptation, backward adaptation, qualitative evaluation, and validation. The MAP employs qualitative and quantitative methods including observation, cognitive and discovery interviews, ideation workshops, prototyping, user testing, co-creation interviews, and psychometric testing. An iterative design is central to the MAP and consistent with both the IQOLA protocol and HCD processes. We will pilot test and execute the MAP to adapt the Patient Reported Outcomes Measurement Information System (PROMIS) Upper Extremity Short Form for use in a mixed literacy hand and upper extremity patient population in Baltimore, Maryland.

### Discussion

The MAP provides an approach for adapting PROMs to a multimedia format. We encourage others to evaluate and test this approach with other questionnaires and patient populations.

**Data Availability Statement:** All relevant data from this study will be made available upon study completion.

**Funding:** This work was funded by the Johns Hopkins Physician Scientist Training Program Microgrant. The funders had no role in study design, data collection and analysis, decision to publish, or preparation of the manuscript.

**Competing interests:** The authors have declared that no competing interests exist.

The development and use of mPROMs has the potential to expand our ability to accurately capture PROs in limited literacy populations and promote equity in PRO measurement.

## Background

Accurately capturing patient-reported outcomes (PROs) in low or limited literacy populations requires patient-reported outcomes measures (PROMs) that are accessible to this population. Although there have been efforts to simplify PROMs, the majority still rely on text elements to convey meaning or question details [1]. Text-based PROMs require basic literacy skills for question comprehension, which in turn precludes illiterate or very limited literacy patients from accurate self-administration [2, 3]. This presents challenges for patients, clinicians, and researchers. Low literacy patients may not understand question meaning, and may experience embarrassment or other barriers in seeking clarification [4]. This may undermine PROM validity due to resulting blank or incomplete questionnaires or answers that do not accurately reflect patient experiences [4, 5]. Because of this, illiterate or low literacy participants may be excluded entirely from research studies that utilize PROMs [1]. Further, health professionals have identified implementation challenges in systems that inadequately support patients with low literacy; these hinder their ability to perform comprehensive evaluations [4].

In the United States, an estimated one-fifth to nearly half of adults have low levels of English literacy [6, 7]. This population has difficulty accessing healthcare [8] and is at higher risk for poor health outcomes [2, 9]. Rates of illiteracy and low literacy are disproportionately higher in ethnic and racial minority populations [6]. While interviewer-administration by clinical or research staff is an alternative mode of administration that allows low literacy patients to self-report, it is labor-intensive and expensive [10]. Interviewer-administration also introduces potential interviewer bias [10] and can cause patient discomfort [4, 8, 11, 12]. Reliance on PROs to understand outcomes and measure healthcare quality risks becoming a systemic means of perpetuating health disparities unless PROMs are designed to be broadly inclusive [13]. For example, excluding low literacy patients from the PROMs development process contributes to inequitable quality improvement practices [13]. These risks are magnified as increasing emphasis on patient-centered care is contributing to wider adoption and routine use of PROMs.

PROMs are routinely administered via two methods: paper-and-pencil or electronically [5]. "Talking Touchscreen" (TT), initially developed for low literacy cancer patients, is an alternative method for administering multimedia PROMs (mPROMs) with audiovisual components [1]. Although other non-PROM questionnaires have been adapted into multimedia formats [3], TT is the only option for PROMs [1]. The Functional Assessment of Cancer Therapy-General (FACT-G) and the Short Form-36 Health Survey (SF-36), when administered using TT, demonstrated good internal consistency; this supports the potential for mPROMs to improve accessibility of PROs in low literacy populations [1, 14–16].

There is currently no consensus approach for developing mPROMs. We present a study protocol to adapt validated, text-based PROMs to mPROMs that can be self-administered in mixed-literacy patient populations. It is our hope that the availability of our Multimedia Adaptation Protocol (MAP) will facilitate the development of future mPROMs and increase their availability in diverse clinical and research settings. This could expand the ability to accurately capture PROs in low literacy populations, providing insights into previously undetected problems and representing an opportunity to improve health equity.

## Methods/Design

### Research ethics

This study has been approved by the MedStar Health Research Institute institutional review board. The identification number is STUDY00002319.

### Developing the Multimedia Adaptation Protocol (MAP)

International Quality of Life Assessment (IQOLA) was one of the first to translate and culturally adapt PROMs into different languages [17, 18]. IQOLA developed a three-stage process that includes 1) translation, 2) psychometric testing of scaling assumptions, and 3) validating and norming of translations [18]. Although this process was initially developed to translate the Short Form Health Survey into cross-culturally comparable versions, it has been used widely in major PROM translation efforts [18]. We chose to use the translation stage of IQOLA's process to guide the development of the MAP because of parallels between the processes of translation and adaptation. The key objective for IQOLA's protocol is to generate instrument versions that are cross-culturally comparable, while the key objective of the MAP is to generate instrument versions that are comparable between modalities. The key metric for success in both protocols is user accessibility and data validity, demonstrated through conceptual equivalence between the source instrument and its translated or adapted versions and reliability of collected data. Because we do not anticipate differences in health concepts between the source instrument and the adapted mPROM, the predominantly deductive approach taken by IQOLA [17] is also appropriate for the MAP.

Our process for developing the MAP began with assembling a multidisciplinary team including two PRO experts, two qualitative research scientists, two social designers, three clinicians, and one biostatistician. Guided by literature review and expert opinion, we identified the critical IQOLA building blocks to construct a conceptual framework. This framework included forward translation, back translation, reconciliation, and validation (Fig 1). Applying this to adapting mPROMs from text-based PROMs, we modeled the four corresponding stages of the MAP: forward adaptation, back adaptation, qualitative evaluation, and validation (green, Fig 1).

We then selected methods to execute each stage. When possible and appropriate, we adopted the previously tested methods in IQOLA's protocol. However, we found that although IQOLA utilizes six different translators to arrive at preliminary and subsequent translated versions [18], there was no direct equivalent of a "translator" to generate candidate versions of mPROMs. For this step, we therefore adopted a human-centered design (HCD) approach. Originating in computer science, engineering, business, and other fields, HCD is increasingly applied in healthcare [19, 20]. HCD, with an emphasis on the end user (i.e., the patient), lends itself particularly well to the patient-centered focus of PRO research [21]. The phases of HCD (empathize, define, ideate, prototype, test) are how the MAP stages are operationalized (pink, Fig 1). We then drew from both the PROs and HCD domains to identify a combination of qualitative and quantitative methods to execute each stage of the MAP (blue, Fig 1). The MAP is represented as three columns in Fig 1 because of the parallel nature of the MAP stages, HCD phases, and methods. For example, reading the MAP horizontally, the MAP forward adaptation stage begins with the HCD phase of empathizing that is carried out with observation and discovery and cognitive interviews.

Finally, extending our community-engaged research approach, we presented an early iteration of the MAP to a research community advisory board to elicit feedback on approach conceptualization and implementation appropriateness. Consultation from this eight-member

## Multimedia Adaptation Protocol

**IQOLA Project's Translation Protocol: A Conceptual Framework**

| Stages | Human-Centered Design Phases | Methods |
|---|---|---|
| **Forward translation** | | |
| **Forward adaptation** | Empathize | **(1) Observation** |
| | | **(2) Discovery and cognitive interviews** |
| | Define | **(3) Analysis** |
| | Ideation | **(4) Ideation workshop and cycles of prototyping and refinement** |
| | Prototype | |
| *mPROM v1.0* | | |
| **Back translation** | | |
| **Back adaptation** | | **(5) Researcher and user testing for conceptual equivalence** |
| *mPROM v2.0* | | |
| **Reconciliation** | | |
| **Qualitative evaluation** | Test | **(6) Consultation with community advisory board** |
| | | **(7) Interviews** |
| *mPROM X.0* | | |
| **Validation** | | |
| **Validation** | | **(8) Psychometric evaluation** |

**Fig 1. Multimedia Adaptation Protocol (MAP).** The IQOLA Project Translation Protocol was used as a guiding conceptual framework to develop the MAP. The parallel components of the MAP include four stages (green), a human-centered design approach (red), and eight distinct methods (blue). (IQOLA = International Quality of Life Assessment; MAP = Multimedia Adaptation Protocol; HCD = human-centered design; mPROM = multimedia patient-reported outcomes measure; v1.0 = version 1.0; v2.0 = version 2.0; X.0 = final version).

board, which comprised a racially and ethnically diverse group of four community members as well as four clinical research experts, aided in the refinement of protocol elements. This board reviewed one iteration of the MAP.

## Study setting and instrument selection

We plan to adapt the Patient Reported Outcomes Measurement Information System (PROMIS) Upper Extremity (UE) Short Form 7a from Item Bank v2.0 [22] to an mPROM in a hand and upper extremity patient population in Baltimore, Maryland. Because the PROMIS UE Item Bank includes questions developed to have known measurement properties (i.e., measure the same dimension and fit the item response theory model on which they were calibrated), any subset of questions from the bank can be used and scored appropriately [23–25]. Although we will begin by using the Short Form, if we identify questions that are not appropriate we can create a customized form from the Item Bank and retain the unidimensional nature of the scale.

Baltimore is a good setting for this study. It is the Maryland county ranked last for both health outcomes [26] and health factors [27] among the state's 24 counties. Baltimore's rate of illiteracy is 15.9% [28] and rate of poverty is 22.4% [29], both above the national average. At the same time, Baltimore has strong health infrastructure, a key asset to the successful implementation of this protocol. This, combined with its greater health services needs relative to the rest of the country, make Baltimore an appropriate setting for a study aimed at improving PRO collection in low literacy patients.

## Multimedia Adaptation Protocol (MAP) methods

The protocol's methods include eight distinct steps (blue, Fig 1). (1) It begins with observation of hand and upper extremity patients, caregivers, and staff in a clinic where patients are routinely asked to complete PROMs. Observation will facilitate understanding the processes of PROM completion and identification of challenges and opportunities in these processes. These findings will inform development of the interview guides used during the next step, (2) cognitive and discovery interviewing. Cognitive interviews offer an opportunity to understand how patients interpret and respond to each item in the source instrument. Semi-structured, discovery interviews elicit a discussion of ease of use, areas of poor design, and perceived barriers and strategies for completion. We will employ purposive sampling of low literacy patients to ensure adequate representation of key study population characteristics in our sample. (3) These data will be analyzed, synthesized, and distilled into key insights that will be conceptualized as design challenges to be used to guide decision-making during the (4) ideation workshop. The workshop will include a structured process with design exercises that facilitate rapid idea generation. These ideas will be produced as several candidate mPROM prototypes (mPROM version 1.0) and undergo refinement to better address the previously identified design challenges. Although the specifics of the prototypes will be determined by the forward adaptation process, components that we anticipate incorporating include audio, visuals, and text. Applying principles of HCD, the prototypes can be static visual representations, thereby decreasing the time and effort needed for prototype production and refinement [30].

(5) These prototypes will undergo back adaptation, during which patients and research personnel blinded to the source instrument translate mPROM prototypes back to text versions. These text back-adaptations are compared to the source instrument; this is done by assessing for conceptual equivalence, defined as the adapted instrument conveying the same content as the source instrument [18]. Nonequivalent questions and responses will be identified and adjusted via consensus by the research team. Prototypes that demonstrate poor conceptual equivalence relative to the others will be discarded.

(6) We will then seek feedback from our community advisory board on the acceptability and usability of mPROM version 2.0. The mPROM will be refined to address issues that are raised by the board before being presented to patients for (7) additional iterative user testing

and refinement. mPROM version 2.0 can be upgraded to a working prototype to optimize the quality and usefulness of data collected during testing. Further co-creation interviews will identify solutions to persistent challenges with the prototype that are not resolved during initial testing.

Central to the MAP, and consistent with both the IQOLA and HCD processes, is its iterative nature. Depending on findings from each step of the MAP, it is possible to undergo additional rounds of interviewing, ideation, prototyping, and user testing. The rigor of the MAP and the robustness of the final mPROM that is generated leverage our phased approach and iteration. Once these processes result in a final mPROM that performs well in both back adaptation and qualitative evaluation, (8) the instrument will undergo validation in the target population via pilot administration and psychometric evaluation. We will defer this final step until a refined, user-tested version of the mPROM has been developed due to the resource and time intensive nature of psychometric evaluation.

## Discussion

Almost all PROMs are based on questions or procedure rendered in text. This intrinsically limits accurate measurement of PROs in low and limited literacy populations when those PROs are obtained via self-administration. We envision a new paradigm in PRO measurement whereby the standard instruments are available in validated, content-equivalent, user-accessible multimedia versions. These instruments are intended to be self-administered and to perform adequately across various levels of patient ability. To facilitate this, we developed the MAP to provide methodology for adapting a validated, text-based PROM to a multimedia version. We believe that this protocol highlights a major inadequacy of existing PROMs and hope that it will facilitate the development of mPROMs, increasing their availability in other medical fields. This work and the MAP therefore represent an opportunity to expand our ability to capture PROs in low or limited literacy populations, to narrow existing health disparities in these populations, and to promote equity in PRO measurement.

The design of the MAP has several strengths. First, its development was guided by a conceptual framework derived from IQOLA's established translation protocol. This lends the MAP a degree of validity and soundness. Second, the MAP includes thorough and iterative assessment of the quality of mPROMs that are being generated. For example, back adaptation allows for assessment of conceptual equivalence and content validity, while user testing allows for assessment of face validity, accessibility, and usability. Finally, the incorporation of HCD into the MAP allows for "translating" PROMs to mPROMs and provides an approach that is efficient (via rapid prototyping), responsive (via user testing), and rigorous (via iteration). It also centers the development process on the patient, who contributes directly or indirectly at every step.

Sharing the MAP with the scientific community allows us to provide a detailed description of the need for the protocol, protocol development processes, and protocol design. This promotes transparency, increases accountability, and facilitates collaboration and coordination. As we pilot test and execute the MAP with the PROMIS UE Short Form in our hand and upper extremity patient population, we invite additional testing and implementation of the MAP in other patient populations. Further research on cost-effectiveness is also needed to evaluate the sustainability of developing mPROMs and administering them on a large scale. Collectively, these efforts can contribute to our limited inventory of mPROMs, reveal ways to improve the protocol, and provide insights to guide implementation.

## Author Contributions

**Conceptualization:** Chao Long, Laura K. Beres, Albert W. Wu, Aviram M. Giladi.

**Methodology:** Chao Long, Laura K. Beres, Aviram M. Giladi.

**Supervision:** Laura K. Beres, Albert W. Wu, Aviram M. Giladi.

**Visualization:** Chao Long.

**Writing – original draft:** Chao Long.

**Writing – review & editing:** Chao Long, Laura K. Beres, Albert W. Wu, Aviram M. Giladi.

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
