## [Decision Letter · Decision Letter 0]

19 Apr 2021

PONE-D-21-02829

Developing a Protocol for Adapting Multimedia Patient-Reported Outcomes Measures for Low Literacy Patients

PLOS ONE

Dear Dr. Long,

Thank you for submitting your manuscript to PLOS ONE. After careful consideration, we feel that it has merit but does not fully meet PLOS ONE’s publication criteria as it currently stands. Therefore, we invite you to submit a revised version of the manuscript that addresses the points raised during the review process.

We look forward to receiving your revised manuscript.

Kind regards,

Frédéric Denis, Ph.D.

Academic Editor

PLOS ONE

Journal Requirements:

Reviewers' comments:

Reviewer's Responses to Questions

**Comments to the Author**

1. Does the manuscript provide a valid rationale for the proposed study, with clearly identified and justified research questions?

Reviewer #1: Partly

Reviewer #2: Yes

2. Is the protocol technically sound and planned in a manner that will lead to a meaningful outcome and allow testing the stated hypotheses?

Reviewer #1: Partly

Reviewer #2: Partly

3. Is the methodology feasible and described in sufficient detail to allow the work to be replicable?

Reviewer #1: No

Reviewer #2: Yes

4. Have the authors described where all data underlying the findings will be made available when the study is complete?

Reviewer #1: Yes

Reviewer #2: Yes

5. Is the manuscript presented in an intelligible fashion and written in standard English?

Reviewer #1: Yes

Reviewer #2: Yes

6. Review Comments to the Author

You may also provide optional suggestions and comments to authors that they might find helpful in planning their study.

Reviewer #1: The Multimedia Adaptation Protocol (MAP) described in this manuscript is innovative and has the potential to improve collection of PROMs in diverse patient populations. It would be helpful to more clearly separate issues of individual question understandability from issues of self-administration. Future sustainability and costs of any multimedia solutions should be briefly discussed. A few additional comments are provided below.

Background:

- Lines 59-60: This phrase is unclear: “...those that incorporate visual elements still rely on text to convey meaning.” How do visual elements rely on text to convey meaning?

- Line 61: This phrase (“patients who are illiterate or have limited literacy may still not be able to complete them”) should be clarified. Do the authors mean that they may not be able self-administer questionnaires if they are only text-based? Interviewer-administration has often been used to help low literacy patients with self-report. It would be helpful to clarify mode (self-administration vs. interviewer-administration) and method (paper, computer, telephone) throughout this manuscript.

- Line 86: The TT is an mPROM method of administration; the method itself does not have “good internal consistency” which is an attribute applicable only to multi-item questionnaires.

Methods/Design:

Lines 160-162: Observation of patients, caregivers, and staff will be helpful to “facilitate understanding the processes of PROM completion and identification of challenges and opportunities in these processes.”

Lines 164-166: The two types of interviews (cognitive and discovery) will be very useful “to understand how patients interpret and respond to each item in the source instrument” and “elicit a discussion of ease of use, areas of poor design, and perceived barriers and strategies for completion.”

Line 175: The technological terms used here (“low-fidelity wireframes or application interfaces”) may not be familiar to most readers.

Lines 179-180: It is not clear how “text back-adaptations” will be “compared to the source instrument to assess for conceptual equivalence.” Conceptual equivalence should also be defined.

Reviewer #2: This paper presents a protocol (Multimedia Adaptation Protocol)for the adaptation of validated patient reported outcomes into a multimedia format to optimize administration for patients with limited health literacy. This paper could be used by other researchers as a template to help decrease disparities in research and I believe would have interest to PLOS ONE readers.

Methods:

Line 99: It would be helpful to describe the 3 states of the IQOLA, since the mPROM protocol is only using one of the stages.

Line 111: how many experts were involved. 1 of each qual, social designers etc? How large was the development team?

Figure 1: On my version I could not find a footnote. There are many acronyms that should be defined in a footnote. It was also hard to know from the figure if all of the 3 columns to the left informed the 4th MAP column and the MAP column is what researchers should do, or if researchers should engage in activities in all 4 columns? Line 132 says it incorporated HCD (pink) into the MAP, but the MAP is the final column. As above, it was hard to know whether column 4 is the final protocol and includes HCD or whether HCD is a separate step, as were the other columns.

Line 137: How large was this board and are any demographic characteristics available? How many iterations of the MAP did they review?

Starting at Line 158 to describe the MAP methods over several paragraphs and the Figure: I think it would help to label the steps listed in this paragraph and have them linked to the box you are referring to in the Figure. As above, I had a hard time understanding how to read and use the figure to adapt PROM. The authors could consider numbering or using letters in the text that then correspond to letters/numbers on the figure so it would be easier to follow the step by step approach to this development.

As this is a methods paper and it should be able to be replicated by these and other researchers, I would have also found a table or a figure with a step by step approach as if not more helpful that the conceptual framework (which as above was still not clear to me what parts were used when and how the IQOLA and HCD were included in the MAP. It was hard to match the text up with the figure.

Line 183: “Identify the prototype that performs the best.” Can the authors describe what that means? I did not see any mention of psychometric properties of the tool. What psychometric properties will be measured and will they define performance?

7. PLOS authors have the option to publish the peer review history of their article (what does this mean?). If published, this will include your full peer review and any attached files.

Reviewer #1: No

Reviewer #2: No

---

## [Author Response · Author response to Decision Letter 0]

25 Apr 2021

Thank you to the editor and the reviewers for their thoughtful comments; we believe that our manuscript is substantially improved thanks to their feedback. Please find a point-by-point response and corresponding revisions in the file, "Response to Reviewers."

---

## [Decision Letter · Decision Letter 1]

12 May 2021

PONE-D-21-02829R1

Developing a protocol for adapting multimedia patient-reported outcomes measures for low literacy patients

PLOS ONE

Dear Dr. Long,

Thank you for submitting your manuscript to PLOS ONE. After careful consideration, we feel that it has merit but does not fully meet PLOS ONE’s publication criteria as it currently stands. Therefore, we invite you to submit a revised version of the manuscript that addresses the points raised during the review process.

We look forward to receiving your revised manuscript.

Kind regards,

Frédéric Denis, Ph.D.

Academic Editor

PLOS ONE

Journal Requirements:

Reviewers' comments:

Reviewer's Responses to Questions

**Comments to the Author**

1. Does the manuscript provide a valid rationale for the proposed study, with clearly identified and justified research questions?

Reviewer #1: Yes

Reviewer #2: Yes

2. Is the protocol technically sound and planned in a manner that will lead to a meaningful outcome and allow testing the stated hypotheses?

Reviewer #1: Yes

Reviewer #2: Yes

3. Is the methodology feasible and described in sufficient detail to allow the work to be replicable?

Reviewer #1: Yes

Reviewer #2: Yes

4. Have the authors described where all data underlying the findings will be made available when the study is complete?

Reviewer #1: Yes

Reviewer #2: Yes

5. Is the manuscript presented in an intelligible fashion and written in standard English?

Reviewer #1: Yes

Reviewer #2: Yes

6. Review Comments to the Author

You may also provide optional suggestions and comments to authors that they might find helpful in planning their study.

Reviewer #1: This is a revised manuscript describing the Multimedia Adaptation Protocol. The revision satisfactorily addressed most, but not all, of the previous review. In addition, the Line numbers mentioned in the authors’ response do not seem to match the manuscript.

This statement should be revised: “Questionnaires administered using TT as the method of administration have good internal consistency…” Internal consistency reliability is a property of a particular questionnaire. It is not accurate to imply that all questionnaires have demonstrated internal consistency reliability, unless the authors have done a comprehensive review of every study that used the TT.

The revised manuscript still uses jargon that may not be familiar to most readers, e.g., “a functional and interactive prototype.”

Reviewer #2: The authors have done a nice job responding appropriately to all of the my and other reviewer comments. This version is much clearer. I appreciate the changes to the text that help match the description to the figure and the simplified figure. I have nothing more to add.

7. PLOS authors have the option to publish the peer review history of their article (what does this mean?). If published, this will include your full peer review and any attached files.

Reviewer #1: No

Reviewer #2: No

---

## [Author Response · Author response to Decision Letter 1]

17 May 2021

Thank you to the editor and reviewers for their comments. We believe that our manuscript is improved thanks to their feedback. Specific responses and changes can be found in the document, "Response to Reviewers."

---

## [Editor Report · Decision Letter 2]

20 May 2021

Developing a protocol for adapting multimedia patient-reported outcomes measures for low literacy patients

PONE-D-21-02829R2

Dear Dr. Long,

We’re pleased to inform you that your manuscript has been judged scientifically suitable for publication and will be formally accepted for publication once it meets all outstanding technical requirements.

Kind regards,

Frédéric Denis, Ph.D.

Academic Editor

PLOS ONE
---

## [Editor Report · Acceptance letter]

27 May 2021

PONE-D-21-02829R2 

Developing a protocol for adapting multimedia patient-reported outcomes measures for low literacy patients 

Dear Dr. Long:

I'm pleased to inform you that your manuscript has been deemed suitable for publication in PLOS ONE. Congratulations! Your manuscript is now with our production department. 

Kind regards, 

on behalf of

Dr. Frédéric Denis 

Academic Editor

PLOS ONE